# Dimethyl fumarate in patients admitted to hospital with COVID-19 (RECOVERY): a randomised, controlled, open-label, platform trial

RECOVERY Collaborative Group*, Peter W. Horby [1,2,60] ✉, Leon Peto[3,4,60], Natalie Staplin[3,5,60], Mark Campbell[3,4], Guilherme Pessoa-Amorim[3], Marion Mafham[3], Jonathan R. Emberson [3,5], Richard Stewart[6], Benjamin Prudon[7], Alison Uriel[8], Christopher A. Green [9], Devesh J. Dhasmana[10,11], Flora Malein [12], Jaydip Majumdar[13], Paul Collini [14], Jack Shurmer[15], Bryan Yates[16], J. Kenneth Baillie [17], Maya H. Buch [18], Jeremy Day [19,20], Saul N. Faust [21], Thomas Jaki [22,23], Katie Jeffery [4,24], Edmund Juszczak [25], Marian Knight[3,26], Wei Shen Lim[25,27], Alan Montgomery[25], Andrew Mumford[28], Kathryn Rowan [29], Guy Thwaites [19,20], Richard Haynes[3,4,5,61] & Martin J. Landray [3,4,5,30,61]

Dimethyl fumarate (DMF) inhibits inflammasome-mediated inflammation and has been proposed as a treatment for patients hospitalised with COVID-19. This randomised, controlled, open-label platform trial (Randomised Evaluation of COVID-19 Therapy [RECOVERY]), is assessing multiple treatments in patients hospitalised for COVID-19 (NCT04381936, ISRCTN50189673). In this assessment of DMF performed at 27 UK hospitals, adults were randomly allocated (1:1) to either usual standard of care alone or usual standard of care plus DMF. The primary outcome was clinical status on day 5 measured on a seven-point ordinal scale. Secondary outcomes were time to sustained improvement in clinical status, time to discharge, day 5 peripheral blood oxygenation, day 5 C-reactive protein, and improvement in day 10 clinical status. Between 2 March 2021 and 18 November 2021, 713 patients were enroled in the DMF evaluation, of whom 356 were randomly allocated to receive usual care plus DMF, and 357 to usual care alone. 95% of patients received corticosteroids as part of routine care. There was no evidence of a beneficial effect of DMF on clinical status at day 5 (common odds ratio of unfavourable outcome 1.12; 95% CI 0.86-1.47; p = 0.40). There was no significant effect of DMF on any secondary outcome.

Severe COVID-19 is characterised by marked inflammation of the lungs, which causes respiratory failure and is usually associated with elevated circulating inflammatory markers such as C-reactive protein (CRP), interleukin 1β (IL-1β), and IL-6[1–4]. This has led to the evaluation of several different kinds of immunomodulation in the treatment of severe COVID-19. Corticosteroids, IL-6 inhibitors, and Janus kinase (JAK) inhibitors have all been found to reduce mortality in hospitalised patients, although the risk of death remains high even when these

A full list of affiliations appears at the end of the paper. *A list of authors and their affiliations appears at the end of the paper.
✉e-mail: recoverytrial@ndph.ox.ac.uk

treatments are used[5–8]. The effectiveness of these drugs proves that inflammation is a modifiable cause of death in patients with COVID-19, and suggests that other ways of modifying the immune response might also be beneficial.

Inflammasomes form part of the innate immune response, and have been proposed as important mediators of COVID-19 lung disease[9,10]. These cytosolic pattern recognition receptor systems stimulate the release of proinflammatory cytokines (in particular IL-1β and IL-18) and activate inflammatory cell death (pyroptosis)[11]. In COVID-19, the degree of inflammasome activation, particularly of the NLR family pyrin domain containing 3 (NLRP3) inflammasome, correlates with disease severity, and analysis of post-mortem lung biopsies from patients with fatal COVID-19 shows that the inflammasome complex is highly activated compared to healthy controls or patients who died from influenza[12,13]. Inflammasome activation was found to be necessary for lung inflammation in a humanised mouse model of SARS-CoV-2 infection, and this effect can be prevented by inflammasome inhibition[14,15]. However, although this pathway has been identified as a promising therapeutic target in humans, treatment with colchicine, which inhibits NLRP3 inflammasome activation, does not improve outcomes in hospitalised patients [16].

Dimethyl fumarate (DMF) is an oral immunomodulator licenced for the treatment of multiple sclerosis and plaque psoriasis. It is a potent inhibitor of inflammasome activation and pyroptosis, and this mechanism is thought to account for its efficacy in preventing multiple sclerosis relapses[17–19]. By inactivating gasdermin D, DMF acts on a common final end point of inflammasome activation, so could inhibit these pathways more effectively than drugs acting on upstream targets or on inflammasome related cytokines[11,17]. In an in-vitro model of SARS-CoV-2 infection, DMF inhibited the production of inflammatory cytokines and also suppressed viral replication, but we are unaware of studies reporting its use for the treatment of COVID-19 in humans[20]. Although generally well tolerated, DMF is often associated with flushing and gastrointestinal symptoms on initiation[21,22].

The UK COVID-19 Therapeutics Advisory Panel (CTAP), a national expert review group set up to identify promising drugs for evaluation, recommended DMF to the RECOVERY chief investigators for investigation among hospitalised patients[23]. Because of limited experience of DMF use in acutely unwell patients and potential tolerability problems, this was planned as an early phase assessment, with subsequent assessment in a larger trial of its effect on mortality if there was evidence of efficacy on surrogate outcomes. Here we report the results of this first randomised assessment of DMF in patients hospitalised with COVID-19, performed as part of the RECOVERY platform trial.

## Results

Between 2 March 2021 and 18 November 2021, 713 (44%) of 1630 patients enrolled into the RECOVERY trial at sites participating in the DMF comparison were eligible to be randomly allocated to DMF (i.e. consent was obtained, DMF was available in the hospital at the time, there was no known indication for or contraindication to DMF, and there was capacity to support the additional follow-up required for this treatment comparison, Fig. 1). Characteristics of eligible and non-eligible patients were similar (Supplementary Table 1). 356 patients were randomly allocated to DMF plus usual standard of care and 357 were randomly allocated to usual standard of care alone. The mean age of study participants in this comparison was 57.1 years (SD 15.7) and the median time since symptom onset was 9 days (IQR 7 to 11 days) (Table 1). At randomisation, 40 (6%) patients did not require oxygen, 535 (75%) required simple oxygen without ventilation, and 135 (19%) required non-invasive ventilation. 674 (95%) were receiving corticosteroids.

Among patients with information on DMF adherence, 306/331 (92%) allocated to DMF received at least one dose, and 248/331 (75%) received at least half of the specified treatment course. Use of other treatments for COVID-19 was similar among patients allocated DMF

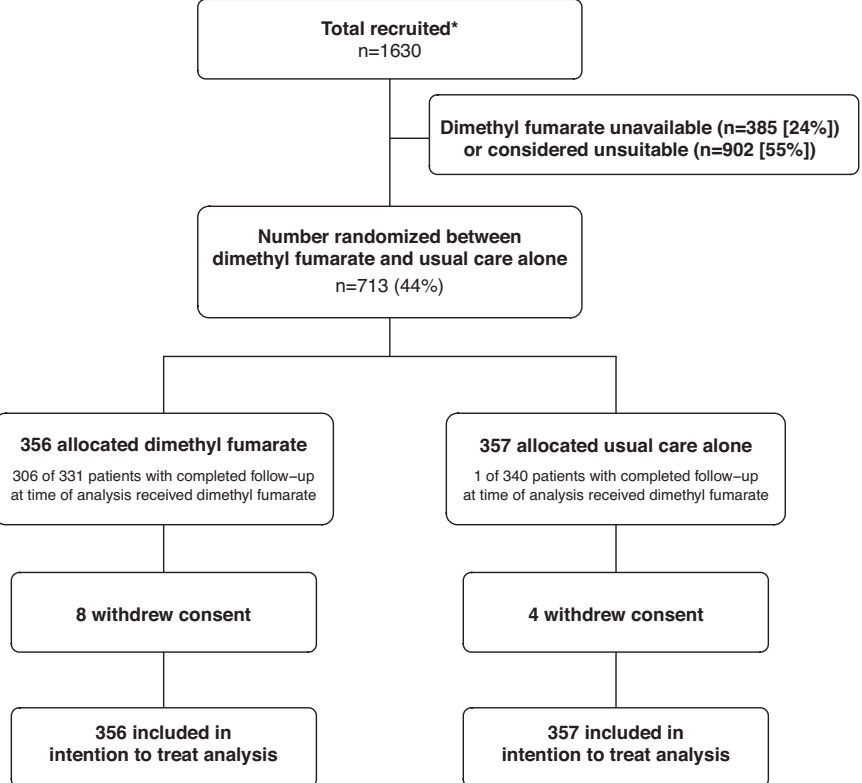

**Fig. 1 | Trial profile.** * Number recruited overall at sites participating in the DMF comparison during the period that adult participants could be recruited. DMF unavailable and DMF unsuitable are not mutually exclusive.

**Table 1 | Baseline characteristics of patients randomised to dimethyl fumarate vs usual care**

| | Dimethyl fumarate (n = 356) | Usual care (n = 357) |
|---|---|---|
| Age, mean years (SD) | 57.5 (16.1) | 56.7 (15.3) |
| ≥18 to <70 | 265 (74%) | 271 (76%) |
| ≥70 to <80 | 61 (17%) | 66 (18%) |
| ≥80 | 30 (8%) | 20 (6%) |
| **Sex** | | |
| Male | 235 (66%) | 246 (69%) |
| Female[a] | 121 (34%) | 111 (31%) |
| **Ethnicity** | | |
| White | 279 (78%) | 282 (79%) |
| BAME[b] | 56 (16%) | 52 (15%) |
| Unknown | 21 (6%) | 23 (6%) |
| Number of days since symptom onset | 9 (7–12) | 9 (7–11) |
| Number of days since hospitalisation | 2 (1–3) | 1 (1–3) |
| Oxygen saturation, % | 94 (92–95) | 94 (92–96) |
| S/F94 | 3.2 (1.2) | 3.1 (1.1) |
| **Ordinal scale** | | |
| 2: Not requiring oxygen or medical care | 0 (0%) | 0 (0%) |
| 3: Requiring medical care but not oxygen | 27 (8%) | 13 (4%) |
| 4: Requiring oxygen without NIV | 257 (72%) | 278 (78%) |
| 5: Requiring oxygen with NIV | 69 (19%) | 66 (18%) |
| 6: Requiring IMV | 3 (1%) | 0 (0%) |
| **Previous diseases** | | |
| Diabetes | 92 (26%) | 89 (25%) |
| Heart disease | 65 (18%) | 49 (14%) |
| Chronic lung disease | 89 (25%) | 71 (20%) |
| Tuberculosis | 0 (0%) | 0 (0%) |
| HIV | 3 (1%) | 1 (<0.5%) |
| Severe liver disease[c] | 3 (1%) | 3 (1%) |
| Severe kidney impairment[d] | 8 (2%) | 8 (2%) |
| Any of the above | 178 (50%) | 161 (45%) |
| **SARS-Cov-2 PCR test result** | | |
| Positive | 334 (94%) | 341 (96%) |
| Negative | 4 (1%) | 2 (1%) |
| Test result not yet known | 18 (5%) | 14 (4%) |
| **Use of steroids** | | |
| Yes | 329 (92%) | 345 (97%) |
| No | 27 (8%) | 12 (3%) |

Mean (SD), median (IQR) or n (%) shown.
[a]No pregnant woman included.
[b]Black and minority ethnic.
[c]Defined as requiring ongoing specialist care.
[d]Defined as estimated glomerular filtration rate <30 mL/min/1.73 m².

and those allocated usual care, including use of baricitinib (44% of participants), and tocilizumab or sarilumab (34% of participants) (Supplementary Table 2).

Primary outcome data are known for 693 (97%) of randomly assigned patients. There was no significant difference between the groups in clinical status at day 5 (common odds ratio of unfavourable outcome 1.12; 95% confidence interval [CI] 0.86–1.47; p = 0.40; Table 2, Figs. 2 and 3).

We found no evidence of an effect of DMF on any secondary or subsidiary outcome (Table 2). There was no significant difference in the time to sustained clinical improvement (rate ratio 0.96; 95% CI 0.80–1.16, p = 0.70) or time to discharge from hospital alive (rate ratio 0.95, 95% CI 0.80–1.13, p = 0.59). At day 5 after randomisation there was no significant difference in S/F$_{94}$ (difference in mean S/F$_{94}$ −0.06; 95% CI −0.22 to 0.10; p = 0.45) or in CRP (difference in geometric mean 2%; 95% CI −18% to 29%; p = 0.84). The proportion of patients with improvement of clinical status by day 10 was similar in both groups (risk ratio 0.95; 95% CI 0.87–1.05; p = 0.31).

Compared to usual care, more participants allocated to DMF suffered flushing (9% vs 3%, risk ratio 2.81; 95% CI 1.44–5.50; p = 0.003) and gastrointestinal symptoms (11% vs 5%, risk ratio 1.99; 95% CI 1.17–3.39; p = 0.01, Table 2), in both cases an excess of around 6% compared to the usual care group. DMF treatment was discontinued because of adverse events in 42 (13%) patients, mainly because of flushing, rash, diarrhoea, or abnormal liver function tests, and 12 (4%) patients required DMF dose reduction (Supplementary Table 3). A further 32 (10%) patients discontinued DMF for reasons other than adverse events, mainly because they were no longer able to take the capsules (Supplementary Table 3). There was one report of a serious adverse reaction believed related to DMF, in a patient whose ALT rose to 5 times the upper limit of normal, although the total number of patients with transaminitis reported was similar in both groups (19% vs 18%, risk ratio 1.05; 95% CI 0.75–1.46; p = 0.78, Table 2). There was no evidence of an effect of DMF on all-cause mortality (13% vs 13%; risk difference 0.6%, 95% CI −4% to 6%) or other safety outcomes, including cause-specific mortality, cardiac arrhythmia, non-coronavirus infections, acute kidney injury, thrombotic events or bleeding events (Table 2, Supplementary Tables 4–6).

## Discussion

In this initial evaluation in the RECOVERY trial, involving over 700 patients hospitalised with COVID-19, treatment with DMF was not associated with improvement in any clinical outcome compared with usual care alone. This is the first randomised trial of DMF for the treatment of COVID-19, and although pre-clinical data suggest that it interferes with inflammatory pathways important to the pathogenesis of COVID-19 pneumonia, this did not translate into any evident benefit of treatment.

Inflammasome activation is a key upstream event in innate immune responses, leading to the release of IL-1β with consequent increase in IL-6 and CRP[24]. DMF blocks a final common pathway of inflammasome signalling in vitro, and in keeping with this, DMF treatment normalises serum IL-1β in patients with multiple sclerosis and reduces serum CRP in an animal model of inflammation[17,25]. At the dose used in RECOVERY it is an effective treatment for relapsing-remitting multiple sclerosis, halving the rate of relapse[20–22]. Inflammasome-mediated signalling is strongly activated in severe COVID-19, which has made it a promising therapeutic target, but colchicine and DMF have both now been evaluated in hospitalised COVID-19 patients and neither has produced a discernible improvement in outcome. The lack of any impact of DMF treatment in RECOVERY, even on serum CRP, suggests this may be because this agent does not block this pathway effectively enough in the context of the COVID-19 inflammatory response, or because activation of this pathway is not causally related to disease trajectory, at least among hospitalised patients receiving current standard treatment. Corticosteroids were received by 95% of the trial population, and a significant proportion also received an IL-6 inhibitor or JAK inhibitor. The lack of benefit of DMF is in contrast to IL-6 and JAK inhibitors, which reduce COVID-19 mortality when used in addition to corticosteroids[7,8]. It remains possible that DMF could have had a beneficial effect in the absence of other immunomodulators, but it appears to add little or nothing to current usual care.

**Table 2 | Effect of allocation to dimethyl fumarate on key study outcomes**

| | Dimethyl fumarate (*n* = 356) | Usual care (*n* = 357) | Treatment effect (95% CI) | *p* value |
|---|---|---|---|---|
| **Primary outcome** | | | | |
| Ordinal scale at day 5[a] | | | | |
| 7 vs 1–6 | 13 (4%) | 10 (3%) | | |
| 6–7 vs 1–5 | 23 (6%) | 22 (6%) | | |
| 5–7 vs 1–4 | 77 (22%) | 73 (20%) | | |
| 4–7 vs 1–3 | 204 (57%) | 193 (54%) | | |
| 3–7 vs 1–2 | 255 (72%) | 249 (70%) | | |
| 2–7 vs 1 | 269 (76%) | 261 (73%) | | |
| Common odds ratio | | | 1.12 (0.86–1.47) | 0.40 |
| **Secondary clinical outcomes** | | | | |
| Sustained improvement in ordinal category within 10 days[b] | 246 (69%) | 258 (72%) | 0.96 (0.80–1.16) | 0.70 |
| Improvement in clinical status at day 10[c] | 246 (69%) | 259 (73%) | 0.95 (0.87–1.05) | 0.31 |
| Baseline-adjusted day 5 S/F94[d] | 3.57 (0.06) | 3.64 (0.06) | −0.07 (−0.23 to 0.09) | 0.38 |
| Baseline-adjusted day 5 CRP[e] | 14.4 (1.2) | 14.0 (1.2) | 2% (−18 to 29%) | 0.84 |
| Median duration of hospitalisation, days | 8 | 8 | | |
| Discharged from hospital alive within 28 days[b] | 274 (77%) | 281 (79%) | 0.95 (0.80–1.13) | 0.59 |
| **Subsidiary clinical outcomes** | | | | |
| Use of ventilation[c,f] | 58/284 (20%) | 60/291 (21%) | 0.99 (0.72–1.37) | 0.95 |
| Non-invasive ventilation | 56/284 (20%) | 56/291 (19%) | 1.02 (0.73–1.43) | 0.89 |
| Invasive mechanical ventilation | 14/284 (5%) | 12/291 (4%) | 1.20 (0.56–2.54) | 0.64 |
| Successful cessation of invasive mechanical ventilation[g] | 0/3 (0%) | 0/0 | - | - |
| Renal dialysis or haemofiltration[c,h] | 7/356 (2%) | 6/355 (2%) | 1.16 (0.39–3.43) | 0.78 |
| **Safety outcomes** | | | | |
| Flushing[c] | | | | |
| Some | 23 (7%) | 11 (3%) | 2.08 (1.03–4.21) | 0.04 |
| Severe | 8 (2%) | 0 (0%) | - | - |
| Subtotal: Any flushing | 31 (9%) | 11 (3%) | 2.81 (1.44–5.50) | 0.0026 |
| Gastrointestinal symptoms[c] | | | | |
| Some | 34 (10%) | 18 (5%) | 1.88 (1.08–3.27) | 0.02 |
| Severe | 4 (1%) | 1 (0%) | 3.99 (0.45–35.51) | 0.21 |
| Subtotal: Any gastrointestinal symptoms | 38 (11%) | 19 (5%) | 1.99 (1.17–3.39) | 0.01 |
| Transaminitis[c] | 57 (19%) | 56 (18%) | 1.05 (0.75–1.46) | 0.78 |
| Acute kidney injury[c] | 9 (3%) | 12 (4%) | 0.75 (0.32–1.75) | 0.51 |
| Non-coronavirus infection[c] | | | | |
| Pneumonia | 18 (5%) | 20 (6%) | 0.90 (0.49–1.68) | 0.75 |
| Urinary tract | 1 (0%) | 4 (1%) | 0.25 (0.03–2.23) | 0.21 |
| Biliary | 0 (0%) | 0 (0%) | - | - |
| Other intra-abdominal | 1 (<0.5%) | 1 (<0.5%) | - | - |
| Blood stream | 4 (1%) | 1 (0%) | 4.01 (0.45–35.71) | 0.21 |
| Skin | 1 (<0.5%) | 1 (<0.5%) | - | - |
| Other | 6 (2%) | 6 (2%) | 1.00 (0.33–3.08) | 1.00 |
| Subtotal: Any non-coronavirus infection | 27 (8%) | 31 (9%) | 0.87 (0.53–1.43) | 0.59 |

All *p* values are 2-sided.

[a]Number of patients with a 'bad' outcome given. Treatment effects are odds ratios for 'bad' vs 'good' outcome. Common odds ratio estimated using a proportional odds model adjusted for ordinal scale at randomisation. For the 22 patients with missing data on ordinal scale at day 5, the median possible category was imputed (rounded up when there are an even number of possibilities).

[b]Treatment effect is a rate ratio estimated using logrank methods.

[c]Treatment effect is a risk ratio.

[d]Treatment effect is difference in mean S/F94 estimated using ANCOVA with adjustment for baseline S/F94. For patients discharged alive by day 5, a value of 4.76 was imputed. All 135 (18.9%) other missing values at day 5 were imputed using multiple imputation.

[e]ANCOVA analyses of log transformed CRP with adjustment for randomisation value were conducted. 276 (38.7%) missing values at day 5 imputed using multiple imputation. Geometric means and approximate standard errors are presented and treatment effect is percentage change in CRP.

[f]Analyses include only those on no ventilation support at randomisation.

[g]Analyses restricted to those on invasive mechanical ventilation at randomisation.

[h]Analyses exclude those on haemodialysis or haemofiltration at randomisation.

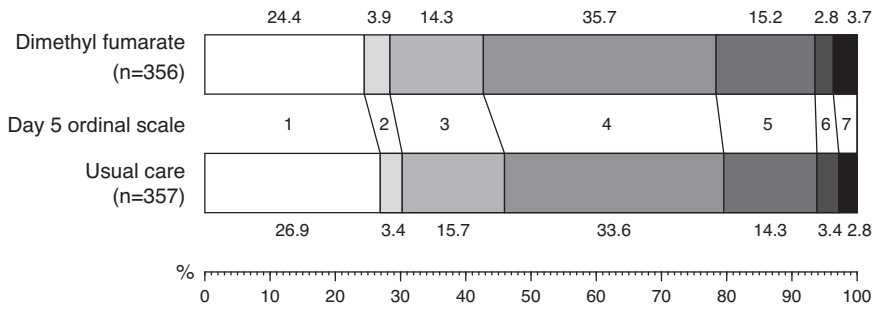

**Fig. 2 | Distribution of clinical ordinal scale at 5 days by randomised allocation.** Percentages of participants in each ordinal category at day 5 are shown.

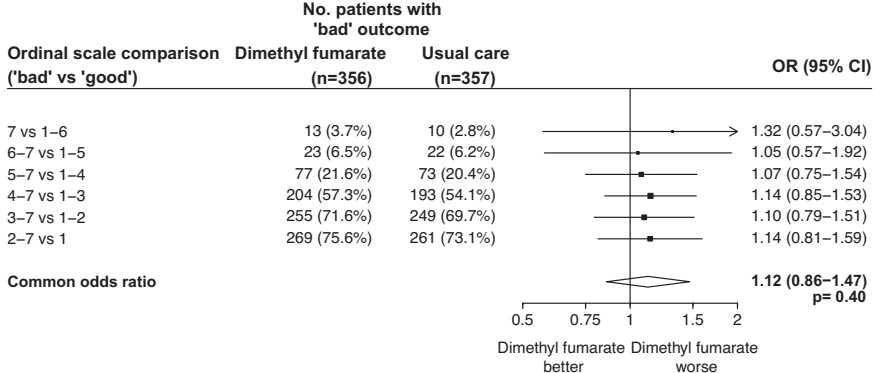

**Fig. 3 | Effects of allocation to dimethyl fumarate on relative odds of a bad outcome on the clinical ordinal scale at day 5, for each alternative definition of bad outcome.** Odds ratio estimates for each ordinal scale comparison are represented by squares (with areas of the squares proportional to the amount of statistical information) and the lines through them correspond to the 95% CIs. Estimates used ordinal logistic regression with adjustment for baseline score. *P* value is 2-sided. OR odds ratio.

Higher doses of DMF could potentially have had a greater anti-inflammatory effect, but may have led to greater problems in tolerability. Treatment was discontinued because of adverse events in 13% of patients, mostly because of flushing, rash, and gastrointestinal symptoms. These are recognised side-effects of DMF, although rarely caused discontinuation in outpatient placebo-controlled trials in patients with multiple sclerosis[21,22]. Other than these adverse effects, no safety concerns of DMF treatment were identified. DMF was discontinued because of abnormal liver function tests in six patients, but ALT elevations are commonly seen in hospitalised patients with COVID and occurred in 18% of participants in the usual care arm[26]. The proportion of patients with transaminitis was similar in the DMF and usual care groups, suggesting DMF was not a significant cause of transaminitis, and highlighting the need for systematic data collection when evaluating adverse events in an open-label study.

Strengths of this trial include that it was randomised, had broad eligibility criteria, and follow up was 97% complete. However, there are some limitations: firstly, as an early phase study, it was not large enough to rule out a benefit in mortality, nor to assess whether treatment effects might have varied among specific groups of patients. Secondly, the trial was open label, so participants and local hospital staff were aware of the assigned treatment. This could potentially affect the assessment of some outcomes, particularly if allocation to DMF led to patients staying in hospital to receive treatment rather than being discharged. However, our protocol specified that treatment was to stop when patients were ready for discharge, and the distribution of clinical status at day 5 provides no evidence to suggest that otherwise healthy patients stayed in hospital to receive DMF (Figs. 2 and 3). Thirdly, although 92% of participants received some DMF, only 75% completed more than half the planned course. In the presence of non-adherence, the intention-to-treat analysis used here would

underestimate any true treatment effect, and a per-protocol analyses would probably introduce bias. Some statistical methods that can provide unbiased estimates of the effect of treatment in the presence of non-adherence, such as inverse probability weighting or g-estimation, require post-randomisation data on prognostic factors that are not available in this trial[27]. Instrumental variable (IV) estimation is an alternative method that does not require these data, and using this approach somewhat increased the uncertainty in the estimated effect of DMF (common odds ratio of unfavourable outcome 1.16; 95% CI 0.81–1.67)[28]. Finally, we only studied patients who had been hospitalised with COVID-19, so do not provide any evidence on the safety and efficacy of DMF in other groups, such as outpatients.

In summary, the results of this randomised trial do not support the use of DMF in adults hospitalised with COVID-19.

## Methods
### Study design and participants

The Randomised Evaluation of COVID-19 therapy (RECOVERY) trial is an investigator-initiated, streamlined, individually randomised, controlled, open-label, platform trial to evaluate the effects of potential treatments in patients hospitalised with COVID-19. Details of the trial design and results for other possible treatments (dexamethasone, hydroxychloroquine, lopinavir-ritonavir, azithromycin, tocilizumab, convalescent plasma, colchicine, aspirin, casirivimab plus imdevimab, and baricitinib) have been published previously[6–8,16,29–34]. The trial is underway at 177 hospital organisations in the United Kingdom supported by the National Institute for Health and Care Research Clinical Research Network, and also at 15 non-UK hospitals (Supplementary Information pp 3-29). Of these, 27 UK hospitals participated in the DMF comparison. The trial is coordinated by the Nuffield Department of Population Health at the University of Oxford (Oxford, UK), the trial

sponsor. The trial is conducted in accordance with the principles of the International Conference on Harmonisation–Good Clinical Practice guidelines and approved by the UK Medicines and Healthcare products Regulatory Agency (MHRA) and the Cambridge East Research Ethics Committee (ref: 20/EE/0101). The protocol and statistical analysis plan are included in the Supplementary Information (pp 63–139) with additional information available on the study website www.recoverytrial.net.

Patients admitted to the hospital were eligible for the study if they had clinically suspected or laboratory confirmed COVID-19 and no medical history that might, in the opinion of the attending clinician, put the patient at significant risk if they were to participate in the trial. Those aged <18 years and pregnant women were not eligible for randomisation to DMF. Written informed consent was obtained from all patients, or a legal representative if patients were too unwell or otherwise unable to provide informed consent. No participant compensation was provided.

## Randomisation and masking
Baseline data were collected using a web-based case report form that included patient demographics, level of respiratory support, major comorbidities, suitability to receive the study treatment, and treatment availability at the study site (Supplementary Information pp 38–40). Contraindications to the DMF comparison were pregnancy, breastfeeding, or known hypersensitivity to DMF, and treatment could not be given parenterally or via feeding tube. Eligible and consenting patients were assigned in a 1:1 ratio to either usual standard of care or usual standard of care plus DMF using web-based simple (unstratified) randomisation with allocation concealed until after randomisation (Supplementary Information pp 36–38). For some patients, DMF was unavailable at the hospital at the time of enrolment or was considered by the managing physician to be either definitely indicated or definitely contraindicated. These patients were not eligible for randomisation between DMF and usual care. Patients allocated DMF were to receive 120mg by mouth every 12 h for the first four doses, followed by 240 mg every 12 h, for total treatment duration of 10 days or until hospital discharge, whichever was sooner. The stepped increase in dose was chosen to minimise flushing and gastrointestinal side effects, and the protocol also allowed dose reduction to a minimum of 120mg once daily if needed to control side effects.

As a platform trial, and in a factorial design, patients could be simultaneously randomised to other treatment groups: i) casirivimab plus imdevimab versus usual care, ii) aspirin versus usual care, iii) baricitinib versus usual care, and iv) empagliflozin versus usual care. Further details of when these factorial randomisations were open are provided in the Supplementary Information (pp 36-38). Participants and local study staff were not masked to the allocated treatment. The trial steering committee, investigators, and all other individuals involved in the trial were masked to aggregated outcome data during the trial.

## Procedures
Participants had daily assessment of clinical status from day 1 to day 10, using a seven-category ordinal scale as follows: 1) discharged alive; 2) in hospital, not requiring oxygen or medical care; 3) in hospital, not requiring oxygen but requiring medical care; 4) in hospital, requiring oxygen via simple face mask or nasal cannula; 5) in hospital, requiring high-flow nasal oxygen or non-invasive ventilation; 6) in hospital, requiring invasive mechanical ventilation or extracorporeal membrane oxygenation; and 7) dead[35]. At baseline and on days 3, 5, and 10, the S/$F_{94}$ ratio was recorded. The S/$F_{94}$ ratio is defined as the ratio of peripheral oxygen saturations ($SpO_2$) to the fraction of inspired oxygen ($FiO_2$), with any supplemental oxygen reduced until $SpO_2$ is <94% (patients were transferred to an oxygen delivery device providing a defined $FiO_2$ if necessary). Details of S/$F_{94}$ measurement and its

rationale are outlined in the Supplementary Information (pp 30, 144–148). Derivation and evaluation of the S/$F_{94}$ endpoint are reported separately[36]. Blood C-reactive protein, creatinine and alanine or aspartate transaminase were measured on days 3, 5, and 10, along with treatment adherence and details of adverse events. The above details were collected into a web-based DMF follow up form developed for this early phase assessment, completed daily until day 10 (Supplementary Information pp 41–45). Another online follow-up form was completed when participants were discharged, had died or at 28 days after randomisation, whichever occurred earliest (Supplementary Information pp 46-53). This recorded information on receipt of other COVID-19 treatments, duration of admission, receipt of respiratory or renal support, and vital status (including cause of death). In addition, routine healthcare and registry data were obtained including information on vital status (with date and cause of death), discharge from hospital, receipt of respiratory support, or renal replacement therapy.

## Outcomes
The primary outcome was clinical status at day 5, as assessed on the ordinal scale. Secondary outcomes were: time to sustained improvement by at least one category on the ordinal scale from baseline (persisting for >1 day), time to discharge from hospital, S/$F_{94}$ ratio at day 5, blood C-reactive protein at day 5, and improvement in clinical status by at least one category at day 10. The initial protocol specified day 5 S/$F_{94}$ as the primary outcome and day 5 clinical status as a secondary outcome, but these were switched in October 2021 when it was realised that discharges before day 5 would lead to significant amounts of missing data for the S/$F_{94}$ outcome. This decision was made by the trial steering committee whilst blinded to the results of the DMF comparison.

Subsidiary clinical outcomes were: use of ventilation and, separately, use of renal dialysis or haemofiltration, among patients not on such treatment at randomisation, and thrombotic events. Pre-specified safety outcomes were: flushing, gastrointestinal symptoms, transaminitis (peak alanine transaminase (ALT) or aspartate transaminase (AST) >3x upper limit of normal), acute kidney injury (peak creatinine >1.5x value at randomisation), cause-specific mortality, bleeding events, major cardiac arrhythmias, and non-coronavirus infections. Information on suspected serious adverse reactions was collected in an expedited fashion to comply with regulatory requirements.

## Statistical analysis
The primary analysis for all outcomes was by intention-to-treat, comparing patients randomised to DMF with concurrent patients randomised to usual care (i.e. those patients who by chance were not allocated DMF, but could have been). For the primary outcome of clinical status at day 5, the common odds ratio of a worse outcome with DMF versus usual care was estimated using ordinal logistic regression with adjustment for baseline score. For 22 participants still alive in hospital on day 5 without a recorded score, the median possible score was imputed. The proportional odds assumption was assessed and there was no evidence that this was violated (p-value from test of proportional odds assumption 0.95).

For time to sustained improvement, the log-rank observed minus expected statistic and its variance were used to test the null hypothesis of equal survival curves (i.e., the log-rank test) and to calculate the one-step estimate of the average rate ratio. Analyses were restricted to the first 10 days as ordinal scores were not collected after this. A similar analysis was used for time to discharge up to day 28, with patients who died in hospital right-censored on day 29. Median time to discharge was derived from Kaplan-Meier estimates.

Comparisons of S/$F_{94}$ ratio and log-transformed CRP at day 5 were performed using analysis of covariance (ANCOVA) adjusted for each participant's baseline value. For patients who were discharged from hospital, for whom it was not possible to measure S/$F_{94}$ ratio at day 5, a

value of 4.76 was imputed (i.e. the maximum value, assuming saturations of 100% when breathing 21% oxygen). Multiple imputation methods were used to account for any other missing data[37]. Risk ratios were used to compare treatment arms for improvement of clinical status at day 10, and for all subsidiary and safety outcomes.

Estimates of rate and risk ratios are shown with 95% confidence intervals. All p-values are 2-sided and are shown without adjustment for multiple testing. The full database is held by the study team, which collected the data from study sites and performed the analyses at the Nuffield Department of Population Health, University of Oxford (Oxford, UK).

It was estimated that enrolment of at least 700 patients would provide 80% power (at $2p = 0.05$) to detect a common odds ratio of 0.67, even if 10% of participants discontinued study treatment before day 5 (based on day 5 outcome frequencies from the existing trial database). Recruitment was halted on 19th November 2021 after target recruitment had been reached. The Trial Steering Committee and all other individuals involved in the trial were masked to outcome data until 28 days after the close of recruitment.

Analyses were performed using SAS version 9.4 and R version 3.4. The trial is registered with ISRCTN (50189673) and clinicaltrials.gov (NCT04381936).

### Role of the funding source

The funder of the study had no role in study design, data collection, data analysis, data interpretation, or writing of the report. D.M.F. was provided from standard National Health Service stocks. The corresponding authors had full access to all the data in the study and had final responsibility for the decision to submit for publication.

### Reporting summary

Further information on research design is available in the Nature Portfolio Reporting Summary linked to this article.

## Data availability

The protocol, consent form, statistical analysis plan, definition & derivation of clinical characteristics & outcomes, training materials, regulatory documents, and other relevant study materials are available online at www.recoverytrial.net. As described in the protocol, the trial Steering Committee will facilitate the use of the study data and approval will not be unreasonably withheld. Deidentified participant data will be made available to bona fide researchers registered with an appropriate institution within 3 months of publication. However, the Steering Committee will need to be satisfied that any proposed publication is of high quality, honours the commitments made to the study participants in the consent documentation and ethical approvals, and is compliant with relevant legal and regulatory requirements (e.g. relating to data protection and privacy). The Steering Committee will have the right to review and comment on any draft manuscripts prior to publication. Data will be made available in line with the policy and procedures described at: https://www.ndph.ox.ac.uk/data-access. Those wishing to request access should apply for access via the Infectious Diseases Data Observatory (https://www.iddo.org/covid19/data-sharing/accessing-data), or complete the form at https://www.ndph.ox.ac.uk/files/about/data_access_enquiry_form_13_6_2019.docx and e-mail to: data.access@ndph.ox.ac.uk.

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

## Acknowledgements
Above all, we would like to thank the patients who participated in this trial. We would also like to thank the many doctors, nurses, pharmacists, other allied health professionals, and research administrators at NHS hospital organisations across the whole of the UK, supported by staff at the National Institute of Health Research (NIHR) Clinical Research Network, NHS Digi-Trials, Public Health England, Department of Health & Social Care, the Intensive Care National Audit & Research Centre, Public Health Scotland, National Records Service of Scotland, the Secure Anonymised Information Linkage (SAIL) at University of Swansea, and the NHS in England, Scotland, Wales and Northern Ireland. The RECOVERY trial is supported by grants to the University of Oxford from UK Research and Innovation (UKRI) and NIHR (MC_PC_19056), the Wellcome Trust (Grant Ref: 222406/Z/20/Z) through the COVID-19 Therapeutics Accelerator, and by core funding provided by the NIHR Oxford Biomedical Research Centre, the Wellcome Trust, the Bill and Melinda Gates Foundation, the Foreign, Commonwealth and Development Office, Health Data Research UK, the Medical Research Council Population Health Research Unit, the NIHR Health Protection Unit in Emerging and Zoonotic Infections, and NIHR Clinical Trials Unit Support Funding. TJ is supported by a grant from UK Medical Research Council (MC_UU_0002/14). WSL is supported by core funding provided by NIHR Nottingham Biomedical Research Centre. Tocilizumab was provided free of charge for this trial by Roche Products Limited. Regeneron Pharmaceuticals supported the trial through provision of casirivimab and imdevimab. The views expressed in this publication are those of the authors and not necessarily those of the NHS, the NIHR, or the UK Department of Health and Social Care. For the purpose of Open Access, the author has applied a CC BY public copyright licence to any Author Accepted Manuscript version arising from this submission. The sponsor was not involved in study design, data collection and analysis or manuscript writing.

## Author contributions
This manuscript was initially drafted by L.P., P.W.H. and M.J.L. All authors contributed to data interpretation and critical review and revision of the manuscript. P.W.H. and M.J.L. vouch for the data and analyses, and for the fidelity of this report to the study protocol and data analysis plan. P.W.H., M.M., J.K.B., M.H.B., J.D., S.N.F., T.J., K.J., E.J., M.K., W.S.L., A.Mo., A.Mu., K.R., G.T., R.H., and M.J.L. designed the trial and study protocol. L.P., M.C., G.P.-A., M.M., R.S., B.P., A.U., C.A.G., D.J.D., F.M., J.M., P.C., J.S., B.Y., and the Data Linkage team at the RECOVERY Coordinating Centre, and the Health Records and Local Clinical Centre staff listed in the Supplementary Information collected the data. N.S. and J.R.E. had access to the study data and did the statistical analysis. PWH and MJL had final responsibility for the decision to submit for publication.

## Funding
UK Research and Innovation (Medical Research Council) and National Institute for Health and Care Research (Grant ref: MC_PC_19056).

## Competing interests
The authors have no conflict of interest or financial relationships relevant to the submitted work to disclose. No form of payment was given to anyone to produce the manuscript. The Nuffield Department of Population Health at the University of Oxford has a staff policy of not accepting honoraria or consultancy fees directly or indirectly from industry (see https://www.ndph.ox.ac.uk/files/about/ndph-independence-of-research-policy-jun-20.pdf).

## Additional information

[1]Pandemic Sciences Institute, Nuffield Department of Medicine, University of Oxford, Oxford, UK. [2]International Severe Acute Respiratory and Emerging Infections Consortium (ISARIC), University of Oxford, Oxford, UK. [3]Nuffield Department of Population Health, University of Oxford, Oxford, UK. [4]Oxford University Hospitals NHS Foundation Trust, Oxford, UK. [5]MRC Population Health Research Unit, University of Oxford, Oxford, UK. [6]Milton Keynes University Hospital NHS Foundation Trust, Milton Keynes, UK. [7]North Tees and Hartlepool NHS Foundation Trust, Stockton-on-Tees, UK. [8]Manchester University NHS Foundation Trust, Manchester, UK. [9]University Hospitals Birmingham NHS Foundation Trust, Birmingham, UK. [10]Victoria Hospital Kirkcaldy, NHS Fife, Kirkcaldy, UK. [11]School of Medicine, University of St Andrews, St Andrews, UK. [12]Liverpool University Hospitals NHS Foundation Trust, Liverpool, UK. [13]Mid Cheshire Hospitals NHS Foundation Trust, Cheshire, UK. [14]Sheffield Teaching Hospitals NHS Foundation Trust, Sheffield, UK. [15]Bolton NHS Foundation Trust, Bolton, UK. [16]Northumbria Healthcare NHS Foundation Trust, Northumberland, UK. [17]Roslin Institute, University of Edinburgh, Edinburgh, UK. [18]Centre for Musculoskeletal Research, University of Manchester, Manchester, UK. [19]Centre for Tropical Medicine and Global Health, Nuffield Department of Medicine, University of Oxford, Oxford, UK. [20]Oxford University Clinical Research Unit, Ho Chi Minh City, Viet Nam. [21]NIHR Southampton Clinical Research Facility and Biomedical Research Centre, University Hospital Southampton NHS Foundation Trust and University of Southampton, Southampton, UK. [22]University of Regensburg, Regensburg, Germany. [23]MRC Biostatistics Unit, University of Cambridge, Cambridge, UK. [24]Radcliffe Department of Medicine, University of Oxford, Oxford, UK. [25]School of Medicine, University of Nottingham, Nottingham, UK. [26]National Perinatal Epidemiology Unit, University of Oxford, Oxford, UK. [27]Respiratory Medicine Department, Nottingham University Hospitals NHS Trust, Nottingham, UK. [28]School of Cellular and Molecular Medicine, University of Bristol, Bristol, UK. [29]Intensive Care National Audit & Research Centre, London, UK. [30]NIHR Oxford Biomedical Research Centre, Oxford University Hospitals NHS Foundation Trust, Oxford, UK. [60]These authors contributed equally: Peter W. Horby, Leon Peto, Natalie Staplin. [61]These authors jointly supervised this work: Richard Haynes, Martin J. Landray. ✉e-mail: recoverytrial@ndph.ox.ac.uk

## RECOVERY Collaborative Group

**Data Monitoring Committee of the RECOVERY Collaborative Group** Peter Sandercock[31], Janet Darbyshire[32], David DeMets[33], Robert Fowler[34], David Lalloo[35], Mohammed Munavvar[36], Adilia Warris[37] & Janet Wittes[38]

**Central Coordinating Office (for the RECOVERY Collaborative Group)** A. Cradduck-Bamford[3], J. Barton[3], A. Basoglu[3], R. Brown[3], W. Brudlo[3], E. Denis[3], L. Fletcher[3], S. Howard[3], K. Taylor[3], G. Cui[3], B. Goodenough[3], A. King[3], M. Lay[3], D. Murray[3], W. Stevens[3], K. Wallendszus[3], R. Wels[3], C. Crichton[3], J. Davies[3], R. Goldacre[3], C. Harper[3], F. Knight[3], M. Nunn[3], H. Salih[3], J. Welch[3], M. Zayed[3], J. Wiles[3], G. Bagley[3], S. Cameron[3], S. Chamberlain[3], B. Farrell[3], H. Freeman[3], A. Kennedy[3], A. Whitehouse[3], S. Wilkinson[3], C. Wood[3], C. Reith[3], K. Davies[3], H. Halls[3], L. Holland[3], R. Truell[3], K. Wilson[3], L. Howie[3], M. Lunn[3], P. Rodgers[3], L. Bowman[3], F. Chen[3], R. Clarke[3], M. Goonasekara[3], W. Herrington[3], P. Judge[3], S. Ng[3], D. Preiss[3], E. Sammons[3] & D. Zhu[3]

**UK National Institute for Health Research Clinical Research Network (for the RECOVERY Collaborative Group)** A. Barnard[39], J. Beety[39], C. Birch[39], M. Brend[39], E. Chambers[39], L. Chappell[39], S. Crawshaw[39], C. Drake[39], H. Duckles-Leech[39], J. Graham[39], T. Harman[39], H. Harper[39], S. Lock[39], K. Lomme[39], N. McMillan[39], I. Nickson[39], U. Ohia[39], E. OKell[39], V. Poustie[39], S. Sam[39], P. Sharratt[39], J. Sheffield[39], H. Slade[39], W. Van't Hoff[39], S. Walker[39], J. Williamson[39], A. De Soyza[39], P. Dimitri[39], S. N. Faust[39], N. Lemoine[39], J. Minton[39], K. Gilmour[39], K. Pearson[39], C. Armah[39], D. Campbell[39], H. Cate[39], A. Priest[39], E. Thomas[39], R. Usher[39], G. Johnson[39], M. Logan[39], S. Pratt[39], A. Price[39], K. Shirley[39], E. Walton[39], P. Williams[39], F. Yelnoorkar[39], J. Hanson[39], H. Membrey[39], L. Gill[39], A. Oliver[39], S. Das[39], S. Murphy[39], M. Sutu[39], J. Collins[39], H. Monaghan[39], A. Unsworth[39], S. Beddows[39], K. Barker-Williams[39], S. Dowling[39], K. Gibbons[39], K. Pine[39], A. Asghar[39], P. Aubrey[39], D. Beaumont-Jewell[39], K. Donaldson[39], T. Skinner[39], J. Luo[39], N. Mguni[39], N. Muzengi[39], R. Pleass[39], E. Wayman[39], A. Coe[39], J. Hicks[39], M. Hough[39], C. Levett[39], A. Potter[39], J. Taylor[39], M. Dolman[39], L. Gerdes[39], C. Hall[39], T. Lockett[39], D. Porter[39], J. Bartholomew[39], L. Dowden[39], C. Rook[39], J. Walters[39], E. Denton[39], H. Tinkler[39], A. Alexander[39], H. Campbell[39], K. Chapman[39], A. Hall[39], A. Rodgers[39], P. Boyle[39], M. Brookes[39], C. Callens[39], H. Duffy[39], C. Green[39], K. Hampshire[39], S. Harrison[39], J. Kirk[39], M. Naz[39], L. Porter[39], P. Ryan[39], J. Shenton[39], J. Warmington[39], M. Amezaga[39], P. Dicks[39], J. Goodwin[39], H. Hodgson[39], S. Jackson[39], M. Odam[39] & D. Williamson[39]

**Health records (for the RECOVERY Collaborative Group)** H. Pinches[40], P. Bowker[40], V. Byrne-Watts[40], G. Chapman[40], G. Coleman[40], J. Gray[40], A. Rees[40], N. Mather[40], T. Denwood[40], D. Harrison[29], G. Turner[41], J. Bruce[42], C. Arkley[43] & S. Rees[43]

**Local Clinical Centre staff (for the RECOVERY Collaborative Group)** J. Alin[6], L. Anguvaa[6], J. Bae[6], G. Bega[6], S. Bowman[6], A. Chakraborty[6], E. Clare[6], S. Fox[6], S. Franklin[6], S. George[6], L. How[6], M. Kennedy[6], J. Mead[6], L. Mew[6], D. Mital[6], L. Moran[6], E. Mwaura[6], M. Nathvani[6], A. Rose[6], D. Scaletta[6], S. Shah[6], L. Siamia[6], O. Spring[6], S. Sutherland[6], F. Teasdale[6], S. Velankar[6], L. Wren[6], F. Wright[6], M. Abouzaid[7], C. Adams[7], A. Al Aaraj[7], O. Alhabsha[7], M. Ali[7], E. Aliberti[7], D. Ashley[7], D. Barker[7], H. Bashir[7], B. Campbell[7], A. Chilvers[7], E. Chinonso[7], V. Collins[7], E. Connell[7], K. Conroy[7], E. Cox[7], J. Deane[7], J. Dunleavy[7], I. Fenner[7], C. Gan[7], I. Garg[7], C. Gibb[7], S. Gowans[7], W. Hartrey[7], F. Hernandez[7], J. Jacob[7], V. Jagannathan[7], V. Jeebun[7], S. Jones[7], M. Khan[7], Y. Koe[7], D. Leitch[7], L. Magnaye[7], T. Mane[7], T. Mazhani[7], N. McDonnell[7],

M. Nafei[7], B. Nelson[7], L. Poole[7], E. Poyner[7], S. Purvis[7], J. Quigley[7], A. Ramshaw[7], H. Reynolds[7], L. Robinson[7], I. Ross[7], R. Salmon[7], L. Shepherd[7], E. Siddle[7], S. Sinclair[7], M. Smith[7], R. Srinivasan[7], K. Stewart[7], R. Taylor[7], G. Wallace[7], S. Wang[7], L. Watson[7], M. Weetman[7], B. Wetherill[7], S. Wild[7], K. Win[7], T. Felton[8], S. Carley[8], R. Lord[8], A. Ustianowski[8], M. Abbas[8], A. Abdul Rasheed[8], T. Abraham[8], S. Aggarwal[8], A. Ahmed[8], A. Ahmed[8], S. Akili[8], P. Alexander[8], A. Allanson[8], B. Al-Sheklly[8], D. Arora[8], M. Avery[8], C. Avram[8], A. Aya[8], J. Banda[8], H. Banks[8], M. Baptist[8], M. Barrera[8], E. Barrow[8], R. Bazaz[8], R. Behrouzi[8], M. Bennett[8], V. Benson[8], A. Bentley[8], A. Bhadi[8], A. Biju[8], A. Bikov[8], K. Birchall[8], S. Blane[8], S. Bokhari[8], P. Bradley[8], J. Bradley-Potts[8], J. Bright[8], R. Brown[8], S. Burgess[8], M. Butt[8], G. Calisti[8], C. Carey[8], N. Chaudhuri[8], S. Chilcott[8], C. Chmiel[8], A. Chrisopoulou[8], E. Church[8], R. Clark[8], J. Clayton-Smith[8], R. Conway[8], E. Cook[8], S. Crasta[8], G. Cummings-Fosong[8], S. Currie[8], H. Dalgleish[8], C. Davies[8], K. Dean[8], A. Desai[8], R. Dhillon[8], J. Digby[8], D. Dolan[8], G. Donohoe[8], A. Duggan[8], B. Duran[8], H. Durrington[8], C. Eades[8], R. Eatough[8], S. Elyoussfi[8], F. Essa[8], G. Evans[8], A. Fairclough[8], D. Faluyi[8], S. Ferguson[8], J. Fielding[8], S. Fiouni[8], J. Flaherty[8], G. Fogarty[8], S. Fowler[8], A. Fox[8], C. Fox[8], B. George[8], V. George[8], S. Giannopoulou[8], R. Gillott[8], A. Gipson[8], S. Glasgow[8], T. Gorsuch[8], G. Grana[8], G. Gray[8], A. Grayson[8], G. Grey[8], B. Griffin[8], J. Guerin[8], P. Hackney[8], B. Hameed[8], I. Hamid[8], S. Hammond[8], S. Handrean[8], A. Harvey[8], J. Henry[8], S. Hey[8], L. Higgins[8], L. Holt[8], A. Horsley[8], L. Howard[8], S. Hughes[8], A. Hulme[8], P. Hulme[8], A. Hussain[8], M. Hyslop[8], J. Ingham[8], O. Ismail[8], A. Jafar[8], R. Jama[8], S. Jamal[8], L. James[8], F. Jennings[8], A. John[8], M. John[8], E. Johnstone[8], D. Kanabar[8], N. Karunaratne[8], Z. Kausar[8], J. Kayappurathu[8], R. Kelly[8], A. Khan[8], W. Khan[8], J. King[8], S. Knight[8], E. Kolakaluri[8], C. Kosmidis[8], E. Kothandaraman[8], S. Krizak[8], K. Kuriakose[8], N. Kyi[8], F. Lalloo[8], G. Lawrence[8], G. Lindergard[8], C. A. Logue[8], L. Macfarlane[8], A. Madden[8], A. Mahaveer[8], L. Manderson[8], G. Margaritopoulos[8], P. Marsden[8], J. Mathews[8], A. Mathioudakis[8], E. McCarthy[8], J. McDermott[8], B. McGrath[8], P. McMaster[8], H. McMullen[8], C. Mendonca[8], A. Metryka[8], D. Micallef[8], A. Mishra[8], H. Mistry[8], S. Mitra[8], S. Moss[8], A. Muazzam[8], D. Mudawi[8], C. Murray[8], M. Naguib[8], S. Naveed[8], P. Ninan[8], M. Nirmalan[8], R. Norton[8], N. Odell[8], R. Osborne[8], G. Padden[8], A. Palacios[8], A. Panes[8], C. Pantin[8], B. Parker[8], L. Peacock[8], A. Peasley[8], N. Phillips[8], M. PI[8], F. Pomery[8], J. Potts[8], N. Power[8], M. Pursell[8], A. Ramchandani[8], A. Rasheed[8], S. Ratcliffe[8], M. Reilly[8], C. Reynard[8], E. Rice[8], M. Rice[8], P. Riley[8], P. Rivera Ortega[8], J. Rogers[8], T. Rogers[8], R. Santosh[8], T. Scoones[8], A. Scott[8], K. Sellers[8], N. Sen[8], T. Shanahan[8], A. Shawcross[8], S. Shibly[8], C. Shilladay[8], A. Simpson[8], S. Sivanadarajah[8], N. Skehan[8], C. Smith[8], J. Smith[8], L. Smith[8], J. Soren[8], W. Spiller[8], K. Stewart[8], J. Stratton[8], A. Stubbs[8], A. Sukumaran[8], K. Swist-Szulik[8], B. Tallon[8], C. Taylor[8], R. Tereszkowski-Kaminski[8], S. Thomas[8], S. Thorpe[8], M. Tohfa[8], R. Tousis[8], T. Turgut[8], M. Varghese[8], G. Varnier[8], I. Venables[8], S. Vinay[8], R. Wang[8], L. Ward[8], C. Warner[8], E. Watson[8], D. Watterson[8], L. Wentworth[8], C. Whitehead[8], D. Wilcock[8], J. Williams[8], E. Willis[8], L. Woodhead[8], S. Worton[8], B. Xavier[8], T. Whitehouse[9], I. Ahmed[9], N. Anderson[9], C. Armstrong[9], A. Bamford[9], H. Bancroft[9], M. Bates[9], M. Bellamy[9], T. Bellamy[9], C. Bergin[9], K. Bhandal[9], E. Butler[9], M. Carmody[9], N. Cianci[9], K. Clay[9], L. Cooper[9], J. Daglish[9], J. Dasgin[9], A. Desai[9], S. Dhani[9], D. Dosanjh[9], E. Forster[9], J. Gresty[9], E. Grobovaite[9], N. Haider[9], B. Hopkins[9], D. Hull[9], Y. Hussain[9], A. Kailey[9], M. Lacson[9], M. Lovell[9], D. Lynch[9], C. McGhee[9], C. McNeill[9], F. Moore[9], A. Nilsson[9], J. Nunnick[9], W. Osborne[9], S. Page[9], D. Parekh[9], C. Prest[9], K. Price[9], V. Price[9], M. Sangombe[9], H. Smith[9], I. Storey[9], L. Thrasyvoulou[9], K. Tsakiridou[9], D. Walsh[9], S. Welch[9], H. Willis[9], L. Wood[9], J. Woodford[9], G. Wooldridge[9], C. Zullo[9], F. Adam[10], K. Aniruddhan[10], J. Boyd[10], P. Cochrane[10], I. Fairbairn[10], S. Finch[10], K. Gray[10], L. Hogg[10], S. Iwanikiw[10], P. Liu[10], J. MacKenzie[10], M. Macmahon[10], I. Murray[10], P. Marks[10], A. Morrow[10], J. Penman[10], A. Pratt[10], J. Ramsay[10], A. Scullion[10], H. Sheridan[10], M. Simpson[10], C. Stewart[10], J. Tait[10], A. Timmins[10], M. Topping[10], P. Hine[12], P. Albert[12], S. Todd[12], I. Welters[12], D. Wootton[12], M. Ahmed[12], R. Ahmed[12], A. Al Balushi[12], M. Anderson[12], A. Atomode[12], P. Banks[12], D. Barr[12], J. Bassett[12], A. Bennett[12], H. Bond[12], A. Bracken[12], T. Brankin-Frisby[12], G. Bretland[12], M. Brodsky[12], J. Brown[12], C. Burston[12], J. Byrne[12], S. Casey[12], L. Chambers[12], D. Coey[12], T. Cross[12], J. Cruise[12], J. Currie[12], L. Dobbie[12], R. Downey[12], A. Du Thinh[12], G. Duncan[12], I. Duru[12], J. Early[12], K. Fenlon[12], I. Fordham[12], H. Frankland[12], S. Glynn[12], J. Goodall[12], S. Gould[12], A. Gureviciute[12], K. Haigh[12], M. Hamilton[12], L. Hampson[12], A. Hanson[12], M. Harrison[12], L. Hawker[12], P. Hazenberg[12], S. Hicks[12], S. Hope[12], M. Howard[12], K. Hunter[12], T. Ingram[12], A. Islim[12], F. Jaime[12], K. Janes[12], B. Johnston[12], S. Kavanagh[12], L. Keogan[12], S. King[12], K. Krasauskas[12], J. Lewis[12], M. Lofthouse[12], P. Lopez[12], C. Lowe[12], Z. Mahmood[12], K. Martin[12], A. Mediana[12], Z. Mellor[12], P. Merron[12], B. Metcalfe[12], M. Middleton[12], K. Monsell[12], N. Nicholas[12], A. Nuttall[12], R. Osanlou[12], L. Pauls[12], R. Price-Eland[12], C. Prince[12], S. Pringle[12], E. Richardson[12], L. Rigby[12], M. Riley[12], A. Rowe[12], E. Rybka[12], M. Samuel[12], D. Scanlon[12], J. Sedano[12], D. Shaw[12], F. Shiham[12], C. Smith[12], S. Stevenson[12], A. Stockdale[12], J. Tempany[12], P. Thu-Ta[12], C. Toohey[12], I. Turner-Bone[12], S. Victoria[12], A. Waite[12], E. Wasson[12], R. Watson[12], V. Waugh[12], R. Westhead[12], L. Wilding[12], K. Williams[12], A. Wood[12], A. Yeoh[12], T. Adeyemo[13], K. Best[13], G. Bridgwood[13], R. Broadhurst[13], C. Brockelsby[13], T. Brockley[13], J. Brown[13], R. Bujazia[13], A. Burton[13], S. Clarke[13], J. Cremona[13], C. Dixon[13], S. Dowson[13], H. Drogan[13], F. Duncan[13], M. Emms[13], H. Farooq[13], D. Fullerton[13], N. G[13], C. Gabriel[13], S. Hammersley[13], R. Hum[13], T. Jones[13], S. Kay[13], E. Kelly[13], M. Kidd[13], D. Lees[13], R. Lowsby[13], D. Maren[13], D. Maseda[13], E. Matovu[13], K. McIntyre[13], H. Moulton[13], K. Nourein[13], K. Pagett[13], A. Ritchings[13], S. Smith[13], J. Taylor[13], K. Thomas[13], K. Turbitt[13], M. Williams[13], S. Yasmin[13], A. Ang[14], J. Belcher[14], M. Boothroyd[14], H. Bowler[14], L. Chapman[14], K. Chin[14], D. Cohen[14], J. Cole[14], H. Colton[14], R. Condliffe[14], M. Cribb[14], S. Curran[14], T. Darton[14], D. de Fonseka[14], T. de Silva[14], A. Dunn[14],

A. Ellwood[14], E. Ferriman[14], H. Foot[14], R. Foster[14], Z. Gabriel[14], J. Greig[14], J. Hall[14], M. Ul Haq[14], S. Hardman[14], E. Headon[14], C. Holden[14], K. Housely[14], A. Howell[14], L. Hunt[14], E. Hurditch[14], F. Ilyas[14], C. Jarman[14], F. Kibutu[14], T. Kitching[14], S. Lassa[14], L. Lewis[14], N. Lewis[14], T. Locke[14], H. Luke[14], A. Lye[14], L. Mair[14], G. Margabanthu[14], P. May[14], J. McNeill[14], S. Megson[14], J. Meiring[14], J. Middle[14], J. Middleton[14], P. Morris[14], T. Newman[14], L. Passby[14], R. Payne[14], M. Plowright[14], A. Raithatha[14], G. Rana[14], S. Renshaw[14], A. Rothman[14], D. Sammut[14], S. Sherwin[14], P. Simpson[14], M. Sterrenburg[14], B. Stone[14], M. Surtees[14], A. Telfer[14], B. Thamu[14], R. Thompson[14], N. Vethanayagam[14], P. Wade[14], S. Walker[14], J. G. R. Watson[14], R. West[14], T. Williams[14], M. Balasubramaniam[15], C. Acton[15], S. Ahmad[15], R. Ahmed[15], A. Ajmi[15], A. Al-Asadi[15], S. Altaf[15], A. Amin[15], A. Bajandouh[15], R. Carey[15], Z. Carrington[15], J. Chadwick[15], S. Cocks[15], C. Dawe[15], S. Farzana[15], O. Froud[15], A. Gibson[15], A. Green[15], P. Hill[15], A. Hindle[15], R. Holmes[15], G. Hughes[15], R. Hull[15], M. Ijaz[15], R. Kalayi[15], M. Khan[15], A. Koirata[15], S. Latham[15], G. Lipscomb[15], K. Lipscomb[15], A. McCorkindale[15], M. McNulty[15], O. Meakin[15], N. Meghani[15], N. Natarajan[15], D. Nethercott[15], P. Nicholas[15], T. Pandya[15], A. Parkinson[15], V. Priyash[15], L. Pugh[15], J. Rafique[15], J. Robertson[15], M. Saleh[15], W. Schneblen[15], B. Sharma[15], O. Sharma[15], D. Shaw[15], Z. Shehata[15], R. Sime[15], S. Singh[15], R. Smith[15], C. Subudhi[15], R. Tallent[15], E. Tanton[15], K. Teasdale[15], D. Tewkesbury[15], P. Thet[15], S. Thornton[15], J. Timerick[15], C. Underwood[15], N. Wang[15], M. Watts[15], I. Webster[15], B. Wilson[15], C. Ashbrook-Raby[16], A. Aujayeb[16], L. Barton[16], J. Bell[16], S. Bourke[16], H. Campbell[16], D. Charlton[16], K. Connelly[16], D. Cooper[16], A. Dawson[16], L. Dismore[16], S. Dodds[16], C. Edwards[16], S. Fearby[16], V. Ferguson[16], A. Green[16], N. Green[16], H. Grover[16], E. Hall[16], I. Hamoodi[16], S. Haney[16], P. Heslop[16], P. Jones[16], H. Lewis[16], J. Luke[16], L. Mackay[16], C. McBrearty[16], G. McCafferty[16], I. McEleavy[16], H. Mckie[16], N. McLarty[16], U. McNelis[16], A. Melville[16], J. Miller[16], A. Morgan[16], S. Parker[16], L. Patterson[16], H. Peggie[16], S. Pick[16], H. Rank[16], D. Ripley[16], S. Robinson[16], E. Rosby[16], J. Rushmer[16], H. Shah[16], T. Smith[16], V. Smith[16], D. Snell[16], J. Steer[16], E. Sykes[16], A. Syndercombe[16], C. Tanney[16], L. Taylor[16], J. Ward[16], R. Warren[16], M. Weatherhead[16], R. Whittle[16], L. Winder[16], M. Ali[44], L. Anderson[44], L. Andrews[44], S. Ashraf[44], D. Ashton[44], G. Babington[44], G. Bartlett[44], D. Batra[44], L. Bendall[44], N. Benetti[44], T. Brear[44], A. Buck[44], G. Bugg[44], J. Butler[44], R. Cammack[44], J. Cantliff[44], L. Clark[44], E. Connor[44], P. Davies[44], M. Dent[44], C. Dobson[44], A. Fatemi[44], M. Fatemi[44], L. Fleming[44], J. Grundy[44], J. Hallas[44], L. Hodgen[44], S. Hodgkinson[44], S. Hodgson[44], L. Howard[44], C. Hutchinson[44], B. Jackson[44], J. Kaur[44], E. Keddie-Gray[44], E. Kendall[44], C. Khurana[44], M. Langley[44], L. Lawless[44], L. Looby[44], M. Meredith[44], L. Morris[44], H. Navarra[44], R. Nicol[44], J. Oliver[44], C. Peters[44], B. Petrova[44], R. Purdy[44], Z. Rose[44], L. Ryan[44], J. Sampson[44], G. Squires[44], J. Squires[44], R. Taylor[44], A. Thomas[44], J. Thornton[44], K. Topham[44], O. Vincent[44], S. Warburton[44], S. Wardle[44], H. Waterfall[44], S. Wei[44], T. Wildsmith[44], L. Wilson[44], R. Sarkar[45], K. Abernethy[45], C. Adams[45], L. Adams[45], A. Addo[45], F. Aliyuda[45], S. Archer[45], A. Arya[45], E. Attubato[45], F. Babatunde[45], M. Bachour[45], P. Balasingam[45], A. Bhandari[45], F. Brokke[45], R. Chauhan[45], V. Chawla[45], R. Chineka[45], A. Davis[45], N. Edmond[45], M. Elbeshy[45], C. Ezenduka[45], S. Ferron[45], C. Gnanalingam[45], D. Gotham[45], M. Hollands[45], M. Iqbal[45], A. Jamal[45], B. Josiah[45], S. Kidney[45], M. Kim[45], K. Koukou[45], T. Kyere-Diabour[45], L. Leach[45], A. Liao[45], A. Maheswaran[45], M. Mansour[45], N. Miah[45], J. Morilla[45], L. Naglik[45], K. Naicker[45], Z. Nurgat[45], S. Rai[45], I. Redknap[45], Z. Rehman[45], A. Ryan[45], Y. Samuel[45], A. Shaibu[45], P. Soor[45], R. Squires[45], W. Stagg[45], W. Ul Hassan[45], P. Vankayalapati[45], E. Vyras[45], A. Williams[45], J. Wood[45], N. Zuhra[45], O. Koch[46], A. Abu-Arafeh[46], E. Allen[46], L. Bagshaw[46], C. Balmforth[46], R. Barnes[46], A. Barnett-Vanes[46], R. Baruah[46], S. Begg[46], S. Blackley[46], M. Braithwaite[46], G. Clark[46], S. Clifford[46], D. Dockrell[46], M. Evans[46], V. Fancois[46], C. Ferguson[46], S. Ferguson[46], N. Freeman[46], E. Gaughan[46], E. Godden[46], S. Hainey[46], R. Harrison[46], B. Hastings[46], S. Htwe[46], A. Ju Wen Kwek[46], M. Ke[46], O. Lloyd[46], C. Mackintosh[46], A. MacRaild[46], W. Mahmood[46], E. Mahony[46], J. McCrae[46], S. Morris[46], C. Mutch[46], S. Nelson[46], K. Nunn[46], D. O. Shea[46], I. Page[46], M. Perry[46], J. Rhodes[46], N. Rodgers[46], J. Schafers[46], A. Shepherd[46], G. Soothill[46], S. Stock[46], R. Sutherland[46], A. Tasiou[46], A. Tufail[46], D. Waters[46], R. Weerakoon[46], T. Wilkinson[46], R. Woodfield[46], J. Wubetu[46], M. Murthy[47], R. Arya[47], A. Baluwala[47], T. Blunt[47], R. Chan[47], L. Connell[47], M. Davey[47], L. Ditchfield[47], G. Drummond[47], A. Ibrahim[47], J. Little[47], N. Marriott[47], B. Mathew[47], M. Moonan[47], T. Nagarajan[47], S. Patel[47], H. Prady[47], L. Roughley[47], S. Sharma[47], H. Whittle[47], G. Hamilton[48], N. Blencowe[48], E. Stratton[48], M. Abraham[48], D. Adams[48], B. Al-Ramadhani[48], B. Amit[48], A. Archer[48], G. Asher[48], G. Aziz[48], A. Balcombe[48], K. Bateman[48], M. Baxter[48], L. Beacham[48], K. Beard[48], K. Belfield[48], N. Bell[48], M. Beresford[48], J. Bernatoniene[48], A. Bhat[48], D. Bhojwani[48], S. Biggs[48], C. Blair[48], J. Blazeby[48], K. Bobruk[48], S. Brooks[48], N. Brown[48], L. Buckley[48], P. Butler[48], A. Cannon[48], C. Caws[48], E. Chakkarapani[48], K. Chatar[48], D. Chatterton[48], B. Chivima[48], E. Clark[48], C. Clemente de la Torre[48], K. Cobain[48], H. Cooke[48], D. Cotterill[48], E. Courtney[48], S. Cowman[48], K. Coy[48], H. Crosby[48], K. Curtis[48], P. Davis[48], O. Drewett[48], K. Druryk[48], R. Duncan[48], H. Dymond[48], K. Edgerley[48], M. Ekoi[48], M. Elokl[48], B. Evans[48], T. Farmery[48], N. Fineman[48], A. Finn[48], L. Gamble[48], F. Garty[48], B. Gibbison[48], L. Gourbault[48], D. Grant[48], K. Gregory[48], M. Griffin[48], R. Groome[48], L. Gurung[48], V. Haile[48], M. Hamdollah-Zadeh[48], A. Hannington[48], R. Harrison[48], J. Heywood[48], A. Hindmarsh[48], N. Holling[48], C. Horrobin[48], R. Houlihan[48], J. Hrycaiczuk[48], H. Hudson[48], K. Hurley[48], J. Iqbal[48], R. Jarvis[48], B. Jeffs[48], A. Jones[48], R. Jones[48], E. King-Oakley[48], E. Kirkham[48], L. Kirkpatrick[48], R. Kumar[48], M. Kurdy[48], A. Lagnado[48], S. Lang[48], L. Leandro[48], H. Legge[48], F. Loro[48], A. Low[48], H. Martin[48], J. Mayer[48], T. Mayo[48], L. McCullagh[48], G. McMahon[48], L. Millett[48], K. Millington[48], J. Mok[48], J. Moon[48], L. Morgan[48], S. Mulligan[48], L. Murray[48], T. Nandwani[48], C. O'Donovan[48], E. Payne[48], C. Penman[48], M. Pezard-Snell[48], J. Pickard[48], M. Pitchford[48],

C. Plumptre[48], D. Putensen[48], A. Ramanan[48], J. Ramirez[48], S. Ratcliffe[48], N. Redman[48], E. Robbins[48], V. Roberts[48], J. Robinson[48], M. Roderick[48], S. Scattergood[48], A. Schadenberg[48], E. Schofield[48], R. Sheppeard[48], C. Shioi[48], J. Shurlock[48], D. Simpson[48], P. Singhal[48], A. Skorko[48], B. Smart[48], N. Smith[48], R. Squires[48], V. Stefania[48], C. Stewart[48], M. Stuttard[48], P. Sugden[48], S. Sundar[48], C. Swanson-Low[48], T. Swart[48], E. Swift[48], A. Tate[48], M. Thake[48], K. Thompson[48], M. Trevelyan[48], K. Turner[48], S. Turner[48], A. Tyer[48], S. Vergnano[48], R. Vincent[48], R. Ward[48], A. White[48], S. Wilkinson[48], J. Williams[48], S. Williams[48], J. Willis[48], H. Winter[48], Z. Woodward[48], L. Woollen[48], R. Wright[48], A. Younes Ibrahim[48], J. Moon[49], R. Baldwin-Jones[49], N. Biswas[49], A. Bowes[49], H. Button[49], E. Cale[49], M. Carnahan[49], E. Crawford[49], E. Damm[49], S. Deshpande[49], D. Donaldson[49], C. Fenton[49], S. Hester[49], Y. Hussain[49], M. Ibrahim[49], S. Islam[49], J. Jones[49], S. Jose[49], H. Millward[49], N. Motherwell[49], J. Nixon[49], S. Pajak[49], S. Passey[49], L. Price[49], M. Rees[49], N. Rowe[49], N. Schunke[49], A. Stephens[49], J. Stickley[49], M. Tadros[49], H. Tivenan[49], A. Gray[50], J. Dear[50], M. Adam[50], R. Al-Shahi Salman[50], A. Anand[50], R. Anderson[50], J. Baillie[50], D. Baird[50], T. Balaskas[50], J. Balfour[50], C. Barclay[50], P. Black[50], C. Blackstock[50], S. Brady[50], R. Buchan[50], R. Campbell[50], J. Carter[50], P. Chapman[50], M. Cherrie[50], C. Cheyne[50], C. Chiswick[50], A. Christides[50], D. Christmas[50], A. Clarke[50], M. Coakley[50], A. Corbishley[50], A. Coull[50], A. Crawford[50], L. Crisp[50], C. Cruickshank[50], D. Cryans[50], M. Dalton[50], K. Dhaliwal[50], M. Docherty[50], R. Dodds[50], L. Donald[50], S. Dummer[50], M. Eddleston[50], S. Ferguson[50], N. Fethers[50], E. Foster[50], R. Frake[50], N. Freeman[50], B. Gallagher[50], E. Gaughan[50], D. Gilliland[50], E. Godden[50], E. Godson[50], J. Grahamslaw[50], A. Grant[50], A. Grant[50], N. Grubb[50], S. Hainey[50], Z. Harding[50], M. Harris[50], M. Harvey[50], D. Henshall[50], S. Hobson[50], N. Hunter[50], Y. Jaly[50], J. Jameson[50], D. Japp[50], H. Khin[50], L. Kitto[50], S. Krupej[50], C. Langoya[50], R. Lawrie[50], A. Levynska[50], M. Lindsay[50], A. Lloyd[50], S. Low[50], B. Lyell[50], D. Lynch[50], J. Macfarlane[50], L. MacInnes[50], I. MacIntyre[50], A. MacRaild[50], M. Marecka[50], A. Marshall[50], M. Martin[50], E. McBride[50], C. McCann[50], F. McCurrach[50], M. McLeish[50], R. Medine[50], H. Milligan[50], E. Moatt[50], W. Morley[50], S. Morrison[50], M. Morrissey[50], K. Murray[50], S. Nelson[50], D. Newby[50], K. Nizam Ud Din[50], R. O'Brien[50], M. Odam[50], E. O'Sullivan[50], R. Penman[50], A. Peterson[50], P. Phelan[50], G. Pickering[50], T. Quinn[50], N. Robertson[50], L. Rooney[50], N. Rowan[50], M. Rowley[50], R. Salman[50], A. Saunderson[50], J. Schafers[50], C. Scott[50], L. Sharp[50], A. Shepherd[50], J. Simpson[50], E. Small[50], P. Stefanowska[50], A. Stevenson[50], S. Stock[50], J. Teasdale[50], E. Thompson[50], J. Thompson[50], I. Walker[50], K. Walker[50], A. Williams[50], N. Wong[51], J. Abrams[51], A. Alkhudhayri[51], N. Aung[51], A. Baldwin[51], O. Bannister[51], J. Barker[51], H. Beddall[51], H. Blamey[51], E. Chan[51], J. Chaplin[51], B. Chisnall[51], C. Cleaver[51], M. Corredera[51], S. Crotty[51], H. Cui[51], B. Davies[51], P. Dey[51], L. Downs[51], S. Gettings[51], B. Hammans[51], S. Jackman[51], P. Jenkins[51], M. Kononen[51], S. Kudsk-Iversen[51], A. Kudzinskas[51], M. Laurenson[51], R. Mancinelli[51], J. Mandeville[51], K. Manso[51], B. Marks[51], S. McLure[51], O. Michalec[51], E. Morgan-Smith[51], A. Ngumo[51], H. Noe[51], R. Oxlade[51], A. Parekh[51], V. Pradhan[51], M. Rahman[51], C. Robertson[51], R. Rule[51], S. Shah[51], H. Smith[51], J. Tebbutt[51], N. Vella[51], M. Veres[51], A. Watson[51], R. West[51], L. Western[51], M. Zammit-Mangion[51], M. Zia[51], G. Cooke[52], L. Young[52], O. Adedeji[52], E. Adewunmi[52], Z. Al-Saadi[52], R. Ashworth[52], J. Barnacle[52], N. Bohnacker[52], A. Cann[52], F. Cheng[52], J. Clark[52], S. Cooray[52], S. Darnell[52], A. Daunt[52], V. Dave[52], A. D'Mello[52], L. Evison[52], S. Fernandez Lopez[52], F. Fitzgerald[52], C. Gale[52], M. Gibani[52], S. Hamilton[52], S. Hunter[52], A. Jimenez Gil[52], S. Johal[52], B. Jones[52], A. Kountourgioti[52], J. Labao[52], V. Latham[52], N. Madeja[52], S. Mashate[52], C. Matthews[52], H. McLachlan[52], A. Mehar[52], J. Millard[52], M. Molina[52], A. Perry[52], S. Rey[52], S. Ryder[52], R. Shah[52], R. Thomas[52], D. Thornton[52], J. Tuff[52], E. Whittaker[52], C. Wignall[52], P. Wilding[52], C. Wong[52], T. Yates[52], C. Yu[52], T. Mahungu[53], H. Tahir[53], A. Abdul[53], R. Abdul-Kadir[53], H. Aboelela[53], M. Al-Khalil[53], N. Allan[53], I. Alshaer[53], M. Anderson[53], M. Araujo[53], G. Badhan[53], A. Bakhai[53], S. Bhagani[53], B. Bobie[53], A. Brraka[53], B. Caplin[53], A. Carroll[53], A. Carroll[53], H. Century[53], E. Cheung[53], D. Cohen[53], O. Coker[53], D. Collier[53], V. Conteh[53], N. Cooper[53], J. Crause[53], N. Davies[53], R. Davies[53], V. Deelchand[53], M. Dosani[53], L. Ehiorobo[53], C. Ellis[53], G. Ferenando[53], J. Franklin[53], P. Gardiner[53], F. Geele[53], J. Gosai[53], N. Handzewniak[53], E. Hanison[53], S. Hanson[53], N. Holdhof[53], H. Hughes[53], C. Jack[53], C. Jarvis[53], V. Jennings[53], H. Koo[53], V. Krishnamurthy[53], A. Kurani[53], Z. Ladan[53], L. Lamb[53], A. Lang[53], V. Le[53], S. Lee[53], S. Lo[53], A. Luintel[53], A. Maharajh[53], H. Mahdi[53], T. Majekdunmi[53], D. Matila[53], S. Melander[53], F. Mellor[53], A. Molloy[53], R. Moores[53], J. Morales[53], G. Moray[53], A. Nandani[53], S. Nasa[53], S. O'Farrell[53], A. Oomatia[53], A. Osadcow[53], J. Osei-Bobie[53], G. Pakou[53], P. Patel[53], C. Patterson[53], E. Pyart[53], E. Quek[53], S. Rabinowicz[53], T. Rampling[53], R. Rankhelawon[53], A. Rodger[53], A. Scobie[53], S. Sharma[53], C. Singh[53], S. Sithiravel[53], T. Sobande[53], P. Talbot[53], P. Taribagil[53], S. Veerasamy[53], G. Wallis[53], J. Whiteley[53], E. Witele[53], A. Wong[53], E. Woodford[53], N. Yaqoob[53], K. McCullough[54], H. Abu[54], C. Beazley[54], H. Blackman[54], P. Bradley[54], D. Burda[54], B. Creagh-Brown[54], J. de Vos[54], S. Donlon[54], C. Everden[54], J. Fisher[54], H. Gale[54], D. Greene[54], O. Hanci[54], L. Harden[54], E. Harrod[54], N. Jeffreys[54], E. Jones[54], J. Jones[54], R. Jordache[54], C. Marriott[54], I. Mayanagao[54], R. Mehra[54], N. Michalak[54], O. Mohamed[54], S. Mtuwa[54], K. Odedra[54], C. Piercy[54], V. Pristopan[54], A. Salberg[54], M. Sanju[54], E. Smith[54], S. Stone[54], E. Tarr[54], J. Verula[54], M. Ainsworth[4], C. Arnison-Newgass[4], A. Bashyal[4], K. Beadon[4], S. Beer[4], A. Bloss[4], L. Buck[4], D. Buttress[4], W. Byrne[4], A. Capp[4], P. Carter[4], L. Carty[4], P. Cicconi[4], R. Corrigan[4], C. Coston[4], L. Cowen[4], N. Davidson[4], K. Dixon[4], L. Downs[4], J. Edwards[4], R. Evans[4], S. Gardiner[4], D. Georgiou[4], A. Gillesen[4], A. Harin[4], M. Havinden-Williams[4], C. Hird[4], A. Hudak[4], P. Hutton[4], R. Irons[4], P. Jastrzebska[4], S. Johnston[4], M. Kamfose[4], K. Lewis[4], T. Lockett[4], F. Maria del Rocio[4], J. Martinez Garrido[4], S. Masih[4], A. Mentzer[4], S. Morris[4], G. Mounce[4], C. O'Callaghan[4], Z. Oliver[4],

J. Patachako[4], S. Paulus[4], E. Perez[4], L. Periyasamy[4], D. Porter[4], S. Prasath[4], C. Purdue[4], M. Ramasamy[4], C. Roehr[4], A. Rudenko[4], V. Sanchez[4], A. Sarfatti[4], M. Segovia[4], T. Sewdin[4], J. Seymour[4], V. Skinner[4], L. Smith[4], A. Sobrino Diaz[4], G. Soni[4], M. Taylor-Siddons[4], H. Thraves[4], C. Tsang[4], M. Vatish[4], Y. Warren[4], E. Wilcock[4], T. Wishlade[4], G. Boehmer[55], A. Alegria[55], R. Kapoor[55], N. Richardson[55], K. Adegoke[55], L. Allen[55], S. Anantapatnaikuni[55], D. Baker[55], E. Beranova[55], H. Blackgrove[55], T. Boumrah[55], P. Christian[55], T. Cosier[55], N. Crisp[55], T. Curtis[55], J. Davis[55], J. Deery[55], A. Elgohary[55], T. Elsefi[55], A. Gillian[55], C. Hargreaves[55], T. Hazelton[55], G. Hector[55], R. Hulbert[55], A. Ionita[55], A. Knight[55], C. Linares[55], S. Liu[55], D. Loader[55], K. Lodhia[55], S. Mandal[55], E. Matisa[55], J. McAndrew[55], K. Mears[55], S. Millington[55], M. Montasser[55], A. Moon[55], C. Oboh[55], P. Offord[55], S. Parashar[55], M. Patel[55], C. Pelham[55], C. Price[55], J. Quindoyos[55], A. Rajasri[55], J. Rand[55], S. Rogers[55], S. Saminathan[55], N. Schumacher[55], A. Skaria[55], R. Solly[55], D. Starnes[55], D. Stephensen[55], S. Stirrup[55], L. Tague[55], S. Tilbey[55], S. Turney[55], V. Vasu[55], A. Velugupati[55], M. Venditti[55], R. Vernall[55], H. Weston[55], Z. Woodward[55], R. Sheridan[56], M. Masoli[56], H. Bakere[56], A. Bowring[56], T. Burden[56], A. Corr[56], P. Czylok[56], L. Dobson[56], A. Forrest[56], E. Goodwin[56], H. Gower[56], A. Hall[56], S. Heddon[56], G. Joseph[56], L. Knowles[56], H. Mabb[56], A. Mackey[56], V. Mariano[56], E. Matkins[56], E. McEvoy[56], L. Mckie[56], P. Mitchelmore[56], L. Morgan[56], T. Nightingale[56], R. Oram[56], C. Oreilly[56], N. Osborne[56], H. Palfrey[56], S. Patten[56], J. Pearce[56], I. Seaton[56], A. Smallridge[56], P. Smith[56], M. Steward[56], D. Sykes[56], J. Tipper[56], S. Todd[56], C. Webb[56], S. Whiteley[56], S. Wilkins[56], N. Withers[56], K. Zaki[56], L. Zitter[56], E. Hunter[57], R. Agbeko[57], A. Bailey[57], K. Baker[57], A. Barr[57], E. Cameron[57], Q. Campbell Hewson[57], A. De Soyza[57], C. Duncan[57], M. Emonts[57], A. Fenn[57], S. Francis[57], J. Glover Bengtsson[57], A. Greenhalgh[57], A. Hanrath[57], H. Hanson[57], C. Hays[57], K. Houghton[57], D. Jerry[57], G. Jones[57], S. Kelly[57], A. Kimber[57], N. Lane[57], J. Macfarlane[57], P. McAlinden[57], I. McCullagh[57], S. McDonald[57], O. Mohammed[57], P. Nwajiugo[57], R. Obukofe[57], J. Parker[57], A. Patience[57], B. Payne[57], R. Percival[57], D. Price[57], Z. Razvi[57], N. Rice[57], S. Robson[57], A. Sanchez Gonzalez[57], B. Shillitoe[57], A. Stanton[57], E. Stephenson[57], N. Trewick[57], S. Tucker[57], R. Welton[57], S. West[57], E. Williams[57], E. Wong[57], F. Yelnoorkar[57], J. Raw[58], R. Tully[58], K. Abdusamad[58], Z. Antonina[58], E. Ayaz[58], B. Blackledge[58], P. Bradley[58], F. Bray[58], M. Bruce[58], E. Bullock[58], C. Carty[58], B. Charles[58], G. Connolly[58], C. Corbett[58], J. Cornwell[58], S. Dermody[58], L. Durrans[58], U. Elenwa[58], E. Falconer[58], J. Flaherty[58], C. Fox[58], J. Guerin[58], D. Hadfield[58], J. Harris[58], J. Haslam[58], S. Hey[58], L. Hoggett[58], A. Horsley[58], C. Houghton[58], L. Howard-Sandy[58], S. Hussain[58], R. Irving[58], P. Jacob[58], D. Johnstone[58], R. Joseph[58], P. Kamath[58], T. Khatun[58], T. Lamb[58], H. Law[58], M. Lazo[58], G. Lindergard[58], S. Lokanathan[58], L. Macfarlane[58], S. Mathen[58], S. McCullough[58], P. McMaster[58], D. McSorland[58], J. Melville[58], B. Mishra[58], G. Moth[58], M. Mulcahy[58], S. Munt[58], J. Naisbitt[58], A. Neal[58], R. Newport[58], G. O'Connor[58], D. O'Riordan[58], I. Page[58], V. Parambil[58], J. Philbin[58], M. Pinjala[58], C. Rishton[58], M. Riste[58], J. Rothwell[58], M. Sam[58], Z. Sarwar[58], L. Scarratt[58], A. Sengupta[58], H. Sharaf[58], J. Shaw[58], J. Shaw[58], K. Shepherd[58], A. Slack[58], D. Symon[58], H. T-Michael[58], A. Ustianowski[58], O. Walton[58], S. Warran[58], S. Williams[58], M. Frise[59], R. Arimoto[59], J. Armistead[59], A. Aslam[59], A. Barrett[59], S. Bartley[59], K. Bostock[59], A. Burman[59], C. Camm[59], R. Carson[59], H. Coles[59], J. D'Costa[59], A. Donohoe[59], E. Duffield[59], F. Emond[59], S. Everden[59], E. Gabbitas[59], E. Garden[59], N. Gould[59], S. Gurung Rai[59], S. Hadfield[59], A. Hayat[59], S. Haysom[59], J. Hilton[59], N. Jacques[59], L. Keating[59], C. Knowles[59], H. Lawrence[59], K. Lennon[59], B. Mitchell[59], T. Okeke[59], S. Rai[59], L. Sathyanarayanan[59], F. Selby[59], M. Thakker[59], S. Vettikumaran[59] & A. Walden[59]

[31]Centre for Clinical Brain Sciences, University of Edinburgh, Edinburgh, UK. [32]University College London, London, UK. [33]School of Medicine and Public Health, University of Wisconsin-Madison, Madison, USA. [34]Department of Medicine and Interdepartmental Division of Critical Care Medicine, University of Toronto, Toronto, Canada. [35]Liverpool School of Tropical Medicine, Liverpool, UK. [36]Lancashire Teaching Hospitals and University of Central Lancashire, Preston, UK. [37]MRC Centre for Medical Mycology, University of Exeter, Exeter, UK. [38]Wittes LLC, Washington, DC, USA. [39]UK National Institute for Health Research Clinical Research Network, London, UK. [40]NHS DigiTrials, Leeds, UK. [41]National Records of Scotland, Edinburgh, UK. [42]Public Health Scotland, Edinburgh, UK. [43]SAIL Databank, University of Swansea, Swansea, UK. [44]Nottingham University Hospitals NHS Trust, Nottingham, UK. [45]Medway NHS Foundation Trust, Gillingham, UK. [46]NHS Lothian: Western General Hospital, Edinburgh, UK. [47]Warrington and Halton Teaching Hospitals NHS Foundation Trust, Warrington, UK. [48]University Hospitals Bristol and Weston NHS Foundation Trust, Bristol, UK. [49]Shrewsbury and Telford Hospital NHS Trust, Shrewsbury, UK. [50]NHS Lothian: Royal Infirmary of Edinburgh, Edinburgh, UK. [51]Buckinghamshire Healthcare NHS Trust, Aylesbury, UK. [52]Imperial College Healthcare NHS Trust, London, UK. [53]Royal Free London NHS Foundation Trust, London, UK. [54]Royal Surrey County Hospital NHS Foundation Trust, Guildford, UK. [55]East Kent Hospitals University NHS Foundation Trust, Canterbury, UK. [56]Royal Devon and Exeter NHS Foundation Trust, Exeter, UK. [57]The Newcastle Upon Tyne Hospitals NHS Foundation Trust, Newcastle, UK. [58]Pennine Acute Hospitals NHS Trust, Salford, UK. [59]Royal Berkshire NHS Foundation Trust, Reading, UK. A full list of members and their affiliations appears in the Supplementary Information.

