## [Peer Review File · Nature Communications]

Dimethyl fumarate in patients admitted to hospital with COVID-19 (RECOVERY): a randomised, controlled, open-label, platform trialEditorial Note: This manuscript has been previously reviewed at another journal that is not operating a transparent peer review scheme. This document only contains reviewer comments and rebuttal letters for versions considered at Nature Communications.

Reviewers' Comments:

Reviewer #2:

Remarks to the Author:

The authors have made significant efforts in addressing previous concerns. However, the high rate of participants who stopped the study early remains a concern. While it is reasonable to use the ITT analysis as the primary analysis, it is important to note that the results from the ITT analysis are likely to favor the null hypothesis due to the dilution of effects caused by participants who did not fully comply. Given the significant percentage of early stoppers, sensitivity analyses using instrumental variable or inverse-probability weighting methods would be beneficial to assess the potential bias introduced by these participants.

Reference:

Hernán MA, Hernández-Díaz S. Beyond the Intent-to-Treat in Comparative Effectiveness Research. Clin Trials. 2012; 9(1): 48-55.

Reviewer #3:

Remarks to the Author:

The authors have responded satisfactorily to my prior comments.

21st September 2023

Reviewer #2:

Remarks to the Author:

The authors have made significant efforts in addressing previous concerns. However, the high rate of participants who stopped the study early remains a concern. While it is reasonable to use the ITT analysis as the primary analysis, it is important to note that the results from the ITT analysis are likely to favor the null hypothesis due to the dilution of effects caused by participants who did not fully comply. Given the significant percentage of early stoppers, sensitivity analyses using instrumental variable or inverse-probability weighting methods would be beneficial to assess the potential bias introduced by these participants.

Reference:

Hernán MA, Hernández-Díaz S. Beyond the Intent-to-Treat in Comparative Effectiveness Research. Clin Trials. 2012; 9(1): 48-55.

RESPONSE: *We agree that this is an issue that warrants further discussion, and have included this as a limitation. We have referred to possible statistical methods that might be used to explore this, and our statistician has performed a sensitivity analysis using an IV estimate. This somewhat increases the uncertainty of the DMF effect estimate, but as this was non-significantly adverse beforehand, it does not substantially affect our conclusions about DMF's lack of efficacy.*

Reviewers' Comments:

Reviewer #2:

Remarks to the Author:

The earlier concerns have been addressed. I have no additional concerns.

Response to reviewer comments 2023-10-13

Reviewer #2 (Remarks to the Author):

The earlier concerns have been addressed. I have no additional concerns

Author: No response required

(this document is just uploaded because not doing so caused submission to fail previously)